# Monoclonal Antibody Disrupts Biofilm Structure and Restores Antibiotic Susceptibility in an Orthopedic Implant Infection Model

**DOI:** 10.3390/antibiotics12101490

**Published:** 2023-09-28

**Authors:** Zachary D. C. Burke, Christopher M. Hart, Benjamin V. Kelley, Zeinab Mamouei, Gideon W. Blumstein, Christopher Hamad, Kellyn Hori, Nicolas Cevallos, Christina Villalpando, Nicole Truong, Amr Turkmani, Micah Ralston, Aaron Kavanaugh, Edgar Tenorio, Lawrence M. Kauvar, Alan Li, Nathanael Prunet, Alexandra I. Stavrakis, Nicholas M. Bernthal

**Affiliations:** 1Department of Orthopedic Surgery, David Geffen School of Medicine, University of California, Los Angeles, CA 90095, USA; hart.christopher2@mayo.edu (C.M.H.); benjamin.v.kelley@uth.tmc.edu (B.V.K.);; 2Department of Orthopaedic Surgery, Cleveland Clinic, 9500 Euclid Avenue, Cleveland, OH 44195, USA; 3Orthopedic Hospital Research Center, David Geffen School of Medicine, University of California, Los Angeles, CA 90095, USAkhori@mednet.ucla.edu (K.H.);; 4Trellis Bioscience, Inc., Redwood City, CA 94063, USA; 5Department of Molecular, Cell and Developmental Biology, University of California, Los Angeles, CA 90095, USA

**Keywords:** biofilm, orthopedic implant infection, *S. aureus*, spinal implant infection, prosthetic joint infection, monoclonal antibody

## Abstract

Bacterial biofilms on orthopedic implants are resistant to the host immune response and to traditional systemic antibiotics. Novel therapies are needed to improve patient outcomes. TRL1068 is a human monoclonal antibody (mAb) against a biofilm anchoring protein. For assessment of this agent in an orthopedic implant infection model, efficacy was measured by reduction in bacterial burden of *Staphylococcus aureus*, the most common pathogen for prosthetic joint infections (PJI). Systemic treatment with the biofilm disrupting mAb TRL1068 in conjunction with vancomycin eradicated *S. aureus* from steel pins implanted in the spine for 26 of 27 mice, significantly more than for vancomycin alone. The mechanism of action was elucidated by two microscopy studies. First, TRL1068 was localized to biofilm using a fluorescent antibody tag. Second, a qualitative effect on biofilm structure was observed using scanning electron microscopy (SEM) to examine steel pins that had been treated in vivo. SEM images of implants retrieved from control mice showed abundant three-dimensional biofilms, whereas those from mice treated with TRL1068 did not. Clinical Significance: TRL1068 binds at high affinity to *S. aureus* biofilms, thereby disrupting the three-dimensional structure and significantly reducing implant CFUs in a well-characterized orthopedic model for which prior tested agents have shown only partial efficacy. TRL1068 represents a promising systemic treatment for orthopedic implant infection.

## 1. Introduction

Biofilms shield bacteria from the immune system and induce a sessile phenotype refractory to antibiotic therapy as compared to the free-growing planktonic state, with biofilm associated bacteria typically being as much as 1000-fold less sensitive to antibiotics. About 70% of clinically significant bacterial infections are biofilm-mediated [1], including infected joint and spine implants, lung infections associated with cystic fibrosis and COPD, osteomyelitis, infective endocarditis, indwelling catheter infections, and chronic non-healing wounds. Clinically, biofilm infections in general are treated with prolonged, high-dose antibiotic treatment, a regimen that increases the potential for toxicity. Since such treatment does not target the biofilm itself, the efficacy may only be temporary. Antibiotic failure in the context of prosthetic joint infections usually requires surgical explantation of the infected metal implant [2]. Although new antibiotics for drug resistant bacteria are in development, few of these efforts are specifically focused on the sessile state. The amelioration of biofilm is being attempted by a variety of indirect methods [3], including aggressive debridement; local delivery of antibiotics to achieve a high concentration at the site of infection; and implant coatings to provide non-specific antimicrobial activity.

Implant-associated infections (IAI) remain a particular challenge in patients undergoing orthopedic and musculoskeletal procedures despite advances in perioperative antibiotics and aseptic surgical technique. In total joint replacement, one of the most common surgical procedures in the United States, periprosthetic joint infection (PJI) occurs in 1–2% of primary replacements [4,5,6,7,8,9,10]. The projected cost of care for PJI in the US by 2030 is over USD 2 billion [8,11,12]. Moreover, PJI confers elevated risk of 5-year mortality [13]. IAI occurs in 2–10% of spine surgery patients, more than doubling the cost of care [14], with implant removal being problematic due to the risk of spine destabilization. Much of the associated morbidity and cost results from formation of bacterial biofilms on the inert implant surfaces. These complex extracellular polymeric substances shield bacteria from the host immune response and increase resistance to systemic antibiotics by 1000-fold [15]. For PJI, published re-infection rates are 7–28% for the “gold-standard” treatment in the US (two-stage surgery to replace the infected components with new counterparts) [16].

There are no approved systemic agents to break down biofilms. Prior preclinical research has demonstrated utility for orthopedic implants of mAbs against bacterial proteins given prophylactically [17], but they have limited breadth of activity against the diverse bacterial species commonly found in prosthetic joint infections. A key structural component of biofilm is extracellular DNA, which is stabilized by bacterial proteins including those in the DNABII family [18]. TRL1068 is a high-affinity (Kd ~50 pM) mAb against a DNABII epitope conserved across both Gram-positive and Gram-negative bacterial species, with efficacy as a treatment demonstrated in rodent models of infective endocarditis and subcutaneous implant infection [19,20]. This mAb was cloned from B cells of a human donor, reducing the potential for toxicity from off-target binding [19]. Surprisingly, antibodies against the target family were found in all twenty healthy (anonymized) humans surveyed, albeit at a low level requiring a sensitive single B-cell assay for detection [21]. However, most such mAbs lack the key favorable features of TRL1068 (broad spectrum epitope specificity and high affinity).

The synergy between TRL1068 and antibiotics has been attributed to the release of bacteria from the biofilm-associated sessile state that is refractory to antibiotics; reversion to the planktonic state restores antibiotic susceptibility. The efficacy of this agent in an orthopedic implant model has not previously been assessed. Addressing this issue, we provide the first evaluation of TRL1068 in an established orthopedic implant model and the first microscopy evidence for biofilm binding and disruption in vivo.

## 2. Results

### 2.1. Ex Vivo Imaging of Fluorescently Tagged TRL1068 Antibody

Methicillin-sensitive *S. aureus* Xen36 is a strong biofilm-forming strain, for which metabolically active cells produce a blue-green bioluminescent signal [22]; use of this strain avoids confounding drug resistance due to biofilm from other mechanisms. Stainless steel pins were inserted into the spine as described in the Methods and visualized by an X-ray (Figure 1). Ex vivo analysis of implants inoculated with Xen36 on POD (post-operative day) 0 and explanted on POD 5, 24 h after tail vein injection of fluorescently tagged antibodies (FL-CAb and FL-TRL1068), demonstrated co-localization of the fluorescent signal with extracellular DNA as demonstrated by TOTO eDNA stain. Only weak background staining was observed for the tagged antibodies and extracellular DNA in the control groups (FL TAG ONLY, FL-CAb, and Vehicle Control) (Figure 2).

### 2.2. Ex Vivo SEM Analysis of Biofilm Three-Dimensional Structure

To assess the qualitative effect of TRL1068 on biofilm structure in isolation, the Day 8 images of explanted pins were taken after TRL1068, CAb, or saline control were given on Days 4 and 7, but before antibiotics were administered. SEM images (Figure 3) of the sterile control implants showed a smooth metal surface with occasional scratches but no three-dimensional structures. By contrast, untreated infected control implants, or implants treated with an isotype control (CAb), had complex, three-dimensional biofilm architecture. Individual *S. aureus* cocci and host cells were observed. Infected implants treated with TRL1068 showed no appreciable three-dimensional biofilm structures and no bacteria in the region of interest; only flattened fibrous regions were observed, which may represent biofilm remnants.

### 2.3. Ex Vivo Implant CFUs

Direct assay of CFUs was assessed at Day 35 (Figure 4), a time point chosen to allow adequate time (14 days) for the biofilm to reform after treatment ceased. The last administration of antibodies was on Day 7, and of vancomycin on Day 21. Viable *S. aureus* Xen36 was found on the implanted pins from only 1 of the 27 animals in the TRL1068+vancomycin group (3.7%), sharply contrasting with the vancomycin only group, with 5/12 (41.7%), and the infected control group, with 8/12 (66.7%). The single implanted pin that yielded bacteria in the TRL1068+vancomycin group grew only 17 CFUs. The efficacy of TRL1068 compared to the untreated and vancomycin controls was statistically significant. Further analysis using Pearson’s Chi-square test found a significant relationship between treatment and infection status (positive or negative) with χ^2^ = 18.7 (df = 3, N = 79), *p* < 0.001. At *p* < 0.05, the TRL1068+vancomycin group had a lower proportion of infected implanted pins compared to the infected control and vancomycin alone groups.

Prior work [19,20] combined with a pharmacokinetics analysis from a single dose rat study established that the serum half-life of TRL1068 in rodents is ~100 h; Cmax for a dose of 15 mg/kg is ~50 µg/mL. The dosing schedule was thus sufficient to maintain serum concentration of the mAb above the in vitro efficacious dose (1.5 µg/mL) for the 7 day treatment period, with washout to below that level over the course of the follow-up period out to Day 35.

### 2.4. Ex Vivo Soft Tissue CFUs

In the peri-implant soft tissues, infection in the untreated group at Day 35 was proportionally higher at *p* < 0.05 (13/13, 100%) compared to the TRL1068+vancomycin (16/27, 59%) and vancomycin (10/12, 83%) groups (Figure 5). Though not significant, a trend was noted for the TRL1068+vancomycin group to have lower mean log10(CFU) (2.7 ± 0.5) compared to the infected control (4.4 ± 0.3, *p* = 0.09) and to the vancomycin alone group (4.2±0.6, *p* = 0.18). Pearson’s Chi-square test showed a significant relation between treatment and infection status, χ^2^ = 8.3 (df = 3, N = 79), *p* = 0.03.

## 3. Discussion

There is an urgent need for less morbid and more effective treatments for orthopedic implant infections, with biofilm formation on the implant surfaces being a key contributor to antibiotic resistance. Currently, there are no approved systemic therapies that target biofilms specifically. A precedent for the present study [17] used a mixture of three mAbs targeting *S. aureus* toxins in a rabbit model for prosthetic joint infection. The mAbs targeted species-specific toxins: alpha-hemolysin; biofilm leukocidins; and clumping factor A. About a 1 log reduction in bacterial load was observed for the mAbs given prophylactically; no post-infection treatment was described. While promising, the species-specific nature of the antigens reduces the utility since about half of clinical cases of prosthetic joint infection include other bacterial species [23]. Disruption of an established biofilm is also of greater utility for integration into current clinical practice.

The data presented here demonstrate the efficacy of a novel anti-biofilm mAb, TRL1068, that has previously shown utility in other animal models, including a Teflon cage implant model. Utility in a validated orthopedic implant infection model is a new result, with novel mechanism of action data supporting the therapeutic hypothesis. The most common pathogen in periprosthetic joint infections, *S. aureus*, was eradicated at study conclusion from implanted pins in 26 of the 27 mice treated with TRL1068+vancomycin, representing a 97% reduction in the mean log10 (CFU) relative to the vancomycin groups and a 96% reduction relative to the infected control group. Importantly, this decline in implant-associated bacteria was achieved by systemic treatment only without any surgical debridement. Moreover, CFUs were enumerated 35 days following bacterial inoculation, which was 28 days after the last dose of TRL1068 and 14 days after the last dose of vancomycin. In light of the previously established pharmacokinetics of TRL1068, the mAb’s concentration was reduced to below the efficacious level by midway through the treatment-free follow-up period. This durable protective effect is a particularly favorable feature, since recurrence of infection following two stage arthroplasty is a clinically significant problem [16].

Fluorescently tagged TRL1068 was shown to bind to the implant-associated biofilm in vivo, with imaging ex vivo. SEM evaluation showed that three-dimensional biofilm structures were abundant on control group implanted pins and absent on those treated with TRL1068. Each result represents a substantive translational and mechanistic advance over prior studies using TRL1068 [19,20]. This native human mAb binds with high affinity (Kd ~50 pM) to a bacterial protein that anchors extracellular DNA (which comprises ~50% of the mass of the biofilm). The epitope is highly conserved across both Gram-positive and Gram-negative bacteria [19], suggesting that TRL1068 may be broadly applicable to periprosthetic joint infections (PJI) resulting from multiple pathogens, although clinical data are required to confirm this hypothesis. The extensive disruption of the biofilm is particularly noteworthy since the eDNA/DNABII component of the biofilm matrix is lower for *S. aureus* than for many other bacterial species [24].

While TRL1068 was highly effective in reducing implant-associated *S. aureus* implant CFUs, an infection of the surrounding soft tissue was only modestly lower than the controls. This may reflect the fact that TRL1068 does not have activity against planktonic bacteria. Non-invasive imaging of *S. aureus* Xen36, which is biased in favor of detecting planktonic bacteria, likewise showed minimal differences in the TRL1068 treated mice compared to controls. The high efficacy of TRL1068 on the implant itself may also reflect a “race to the surface” [25,26], which refers to competition at the implant surface between host–cell integration and bacterial colonization. That is, a higher bacterial load is required to establish an implant infection for a delayed bacterial inoculation versus an acute infection [26]. Further investigation of the kinetics of biofilm disruption in vivo will therefore be of interest. Optimization of antibiotic dose regimen to deal with the planktonic bacteria released from the biofilm also warrants attention, as clinical practice includes pathogen-specific antibiotics [27,28,29].

## 4. Materials and Methods

### 4.1. Mice

Twelve-week-old, 20–25 g C57BL/6 wild-type mice (Jackson Laboratory, Bar Harbor, ME, USA) were housed 4 per cage under a 12 h light and dark cycle with free access to water and a standard pellet diet. Institutional veterinary staff evaluated the health of all mice daily. A total of 85 mice were used. All animal procedures were approved by the UCLA Animal Research Committee (ARC 2012-104-03J).

### 4.2. Monoclonal Antibodies (mAbs)

TRL1068 (20 mg/mL) and an IgG1 isotype control mAb (designated as CAb) known to be non-reactive with mammalian proteins (7.4 mg/mL) were provided by Trellis Bioscience, Inc. (Redwood City, CA, USA).

### 4.3. Antibiotics

Vancomycin purchased from American Pharmaceutical Partners Inc. (Los Angeles, CA, USA) was reconstituted as recommended by the manufacturer.

### 4.4. Staphylococcus aureus Bioluminescent Strain

Methicillin-Sensitive *Staphylococcus aureus* Xen36 (PerkinElmer; Hopkinton, MA, USA) is derived from clinical isolate ATCC 49525. It contains a bioluminescent lux operon that produces a blue-green luminescent signal in viable, metabolically active cells [22,30,31], and it carries a kanamycin-resistance marker. Frozen stock was streaked onto agar plates (Luria broth supplemented with 1.5% bacto agar; Teknova; Hollister, CA, USA) containing 200 µg/mL kanamycin (Sigma-Aldrich, St Louis, MO, USA) and incubated for 24 h at 37 °C. Single colonies exhibiting bioluminescence were selected and subcultured, yielding mid-logarithmic phase cells for the studies.

### 4.5. Imaging

In vivo bioluminescence signals from Xen36 were measured using the IVIS system (Perkin Elmer, Hopkinton, MA, USA). Ex vivo analysis of stained implants used Imaris software v.9.9 (Oxford Instruments, Enkoping, Sweden) with light intensity measured in lumens/m^2^. The excitation/emission for the antibody fluorescent tag VivoTag^®^ 680XL (Perkin Elmer, Hopkinton, MA, USA) [32] has minimal overlap with that of TOTO^®^-1 (Thermo Fisher Scientific, Waltham, MA, USA), a cell-impermeant DNA-binding fluor that has been shown to reliably stain extracellular DNA [33]. Fluorophores were imaged sequentially; TOTO^®^-1 was excited with a 514 nm laser, 0.4% laser power, and fluorescence collected from 519-628 nm; VivoTag^®^ 680XL was excited with a 633 nm laser, 24% laser power, and fluorescence collected from 661-759 nm. Z-stacks were acquired on a Zeiss LSM880 confocal microscope using a 10X Plan-Apochromat M27 objective with a 0.45 numerical aperture.

### 4.6. In Vivo Spine Surgery Model

Spinal implant infection was performed as previously described [34]. Briefly, a midline skin incision was centered over the lower lumbar spine. The fascia was incised and the para-spinal muscles sharply dissected away from their osseous attachments to gain access to the posterior elements of the L4 vertebra. A sterile 0.1mm diameter L-shaped stainless-steel pin was then passed through the L4 spinous process of the lumbar spine. The short arm of the pin was placed intra-osseously, and the long arm was left immediately adjacent to the posterior elements of the spine. X-rays were taken to confirm pin placement (Figure 1). Following placement of the pin, 1 × 10^3^ CFU of *S. aureus* Xen36 in 2 mL saline, or sterile control, was inoculated directly onto the implant. The wound was closed in a layered fashion including an independent fascial layer and skin layer.

### 4.7. Antibody and Antibiotic Administration

Mice were randomized into four experimental groups: (i) sterile control (n = 6); (ii) infected control (n = 12); (iii) vancomycin alone (n = 12); (iv) TRL1068+vancomycin (n = 27). For the mechanism of action microscopy studies, an additional isotype control antibody (CAb) was included. After 4 days to establish an *S. aureus* Xen36 biofilm-associated implant infection, subcutaneous injections were given into the inguinal area: mAbs at 15 mg/kg on Days 4 and 7 [19]; vancomycin at 120 mg/kg every 12 h from Days 7–21. Mice in the control groups received saline injections. Injections were performed sequentially with one cage of animals from each group to minimize confounding. Prior work [19,20] had established that the indicated mAb dosing schedule assures serum concentration that is well above the in vitro efficacious concentration of 1.5 µg/mL for 10–14 days.

### 4.8. Ex Vivo Staining of Biofilm

Animals (n = 3 per group) were implanted and inoculated with *S. aureus* Xen36 on POD0 and tail vein injected with fluorescently tagged TRL1068 (Fl-TRL1068) at 15 mg/kg, fluorescently tagged isotype IgG control antibody (Fl-CAb) at 15 mg/kg, fluorescent tag alone, or vehicle control on POD4. Animals were sacrificed 24 h following administration of the mAbs or controls, implants were retrieved and gently washed with 1X phosphate buffered saline (PBS), stained with TOTO^®^-1, and imaged by confocal microscopy.

### 4.9. Ex Vivo Scanning Electron Microscopy of Implants

Animals were sacrificed on Day 8 (n = 3 per group) and implants retrieved and imaged with a Scanning Electron Microscope (SEM; Zeiss Supra-40 VP Field Emission). At the time of implantation, the short end of the 10 mm L-shaped pin was inserted into the L4 spinous process. To avoid damage to the biofilm on the long end of the explanted pins, the short end of the pin was carefully held with forceps and removed from the bone. The explanted pins were washed in PBS and then fixed in 2.5% glutaraldehyde for 48 h. The pins were post-fixed in 2.0% osmium tetroxide for 1 h, rinsed in PBS, and dehydrated through a graded series of ethanol to 100% [33,35,36]. The pins were laid down on conductive tape for imaging and, therefore, only one side of the pins was imaged. Only the long end of the pins was imaged. Magnifications were standardized between groups, as was the viewing angle to the extent possible.

### 4.10. Ex Vivo Soft Tissue and Implant CFUs

Implanted pins and surrounding tissue were processed separately. Following sacrifice of mice on Day 35, implanted pins were removed and adherent bacteria detached by sonication in 500 µL of 0.3% Tween-polysorbate 80 in TSB for 15 min. Additionally, 0.1–0.2 g of the bone and soft tissue immediately adjacent to the implant was harvested and homogenized (Pro200H Series homogenizer; PRO Scientific, Oxford, CT, USA). Bacteria from the implant and tissue were quantified after overnight culture on agar plates as CFUs/mL and CFUs/g, respectively, and confirmed to be *S. aureus* Xen36 by bioluminescent imaging.

### 4.11. Statistical Analysis

Experimental group sizes were based on previous experiments [31]. A minimum of ten mice surviving to experiment completion were needed for control groups, with treated groups double in size. For qualitative microscopic experiments, at least three replicates were used for each group. For statistical analysis and plotting, significance of differences among the treatment groups used one-way analysis of variance (ANOVA) followed by Tukey’s HSD (honest significant difference) post hoc test. To further analyze treatment effect on implant or tissue infection, CFUs ≥ 1.0 were classified as positive and CFUs < 1.0 as negative, and a Pearson’s chi-square test of independence was performed, followed by a z-test for independent infection status proportions per group. Statistical analyses were performed using IBM SPSS Statistics version 20 (IBM, Armonk, NY, USA).

## 5. Conclusions

The results reported here support the use of TRL1068 as a clinical candidate to treat orthopedic implant infection, demonstrating near complete eradication of implant-associated bacteria in a validated mouse orthopedic implant infection model. In particular, the ability of systemically administered mAb to reach the extravascular implant-associated biofilm provides a compelling argument for clinical evaluation of the mAb as an addition to current standard of care.

Despite decades of research and advances in perioperative antibiotics, aseptic surgical techniques, and patient optimization procedures, implant-associated infections remain a major cause of morbidity and mortality [2]. Although interest in breaking down biofilms has been increasing, TRL1068 is the first clinical candidate specifically focused on eliminating the biofilm itself rather than the bacterial pathogens that are protected in biofilm. A key advantage of this approach is that new antibiotics are typically reserved as the last line of defense to avoid selecting for escape mutants. By contrast, the potential for bacteria to escape from TRL1068 is expected to be low, enabling front line use. Specifically, the target protein is only exposed to the mAb after the producing cell is already dead, precluding a growth advantage for mutants that escape binding by TRL1068.

The present study’s ex vivo SEM analysis of implant-associated biofilm structure after TRL1068 treatment provides the first images in the literature that directly demonstrate the effect of TRL1068 on biofilm structure in vivo, without concurrent antibiotics. These images provide qualitative corroboration of the significant quantitative differences seen in POD35 mean implant CFUs between the TRL1068+vancomycin and other experimental groups. The complete absence of three-dimensional biofilm structures on the implants treated with TRL1068 contrasts starkly to the robust structures observed in the control groups. This finding supports the hypothesis that TRL1068 potentiates the effect of antibiotics by degrading the three-dimensional structure of biofilms, thus releasing bacteria into a planktonic, and therefore antibiotic susceptible, state.

## Figures and Tables

**Figure 1 antibiotics-12-01490-f001:**
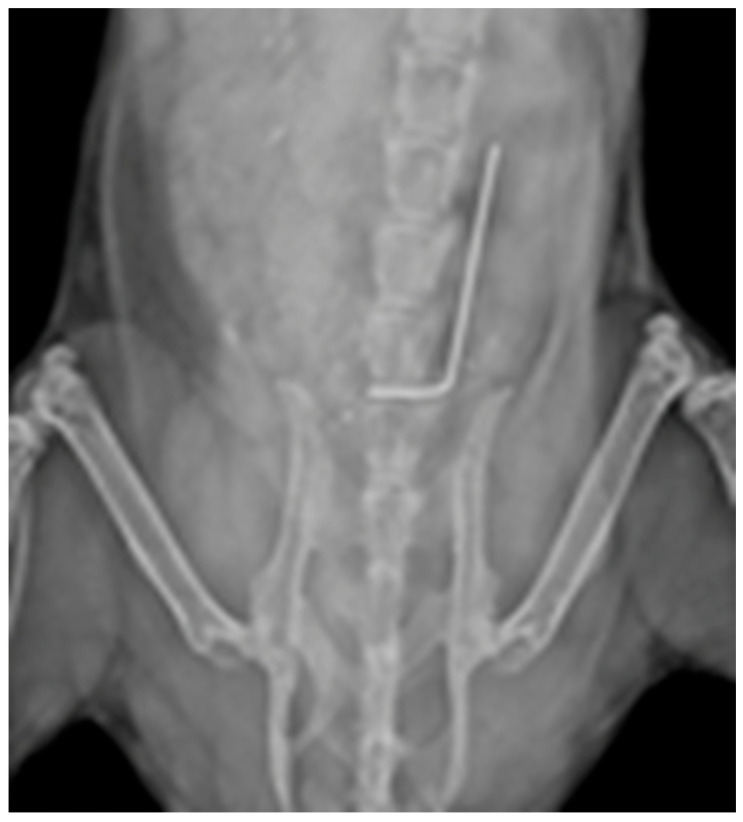
X-ray of spinal implant in situ. A 0.1 mm diameter, 10 mm long stainless-steel pin, with a 90-degree bend 2 mm from the proximal end, was implanted into the L4 spinous process.

**Figure 2 antibiotics-12-01490-f002:**
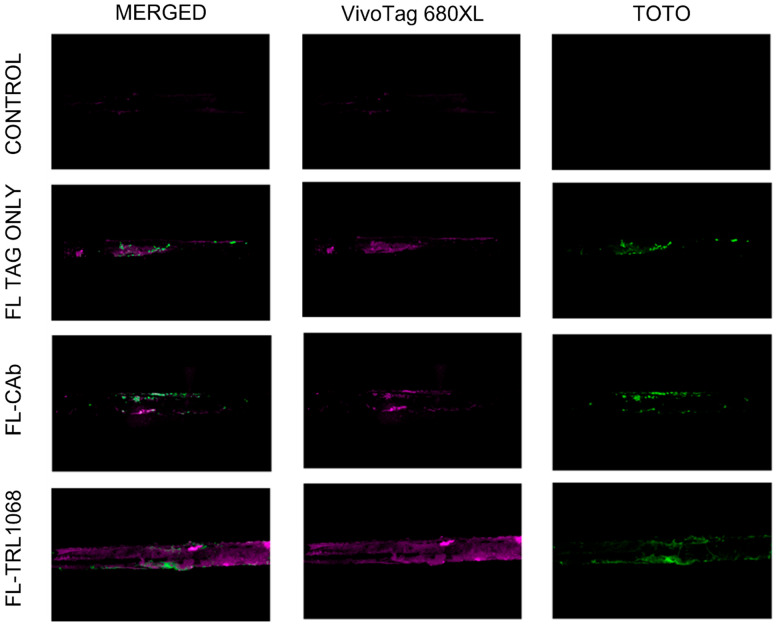
Ex vivo localization of fluorescently tagged TRL1068. The in vivo spinal implant infection model was used. Isotype IgG control antibody and TRL1068 were tagged with a fluorescent dye using the VivoTag^®^ 680XL kit. Animals were implanted and inoculated with *S. aureus* Xen36 on POD0 and received fluorescently tagged TRL1068 (Fl-TRL1068), fluorescently tagged isotype IgG control antibody (Fl-CAb), fluorescent tag alone, or vehicle control on POD4. Twenty-four hours after antibody dosing, the animals were sacrificed, and the implants were harvested and stained with TOTO^®^-1 eDNA stain to co-localize bacterial biofilm. Images were taken with confocal microscopy. CONTROL (bacterial biofilm alone, vehicle control, no dyes), FL-TAG ONLY (Infected control + dyes alone (VivoTag^®^ 680XL alone and TOTO^®^-1 eDNA stain)), FL-CAb (biofilm-infected implant + IgG isotype control antibody tagged with VivoTag^®^ 680XL + TOTO^®^-1 eDNA stain), and FL-TRL1068 (biofilm infected implant + TRL1068 antibody tagged with VivoTag^®^ 680XL + TOTO^®^-1 eDNA stain) groups are shown here. There was no fluorescent signal in group A. Biofilm staining as measured by TOTO^®^-1 signal intensity was similar in groups B, C, and D (1.56 × 10^7^ lumens/m^2^, 2.32 × 10^7^ lumens/m^2^, and 1.07 × 10^7^ lumens/m^2^, respectively). By comparison, there was a seven-times greater signal intensity within the VivoTag^®^ 680XL wavelength in group D (FL-TRL1068) relative to control groups B and C (7.19 × 10^7^ lumens/m^2^ vs. 1.45 × 10^7^ lumens/m^2^ and 1.48 ×10^7^ lumens/m^2^, respectively). This confirms localization of FL-TRL1068 to biofilm in vivo.

**Figure 3 antibiotics-12-01490-f003:**
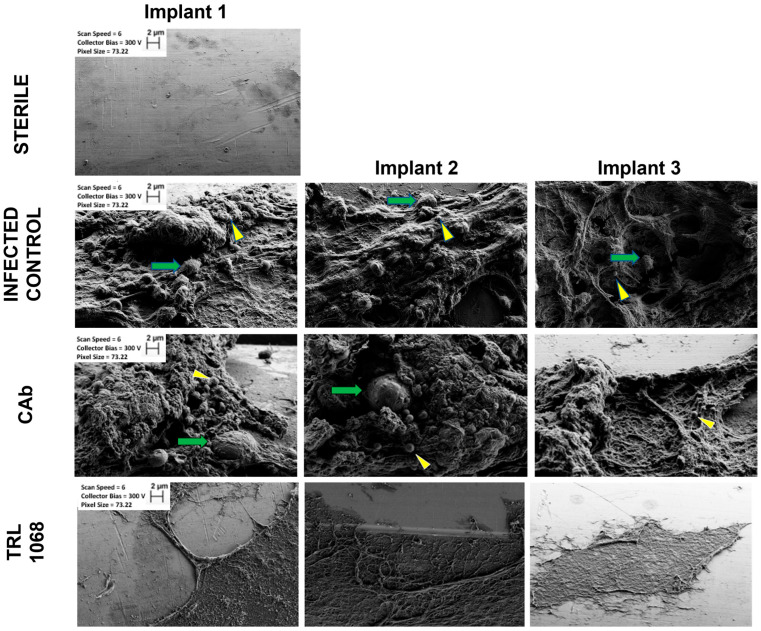
Ex vivo scanning electron microscopy (SEM) evaluation of three-dimensional biofilm structure. Stainless steel pins were implanted into the L4 spinous process of mice on Day 0 and inoculated with saline (sterile control) or 1× 10^3^
*S. aureus* Xen36. On Day 8, the animals were sacrificed (n = 3 per group), and the pins were imaged according to a standard SEM protocol. Experimental groups: (Sterile) sterile control showing a smooth metal pin; (Infected Control) infected control (bacteria alone) showing a thick 3-dimensional biofilm structure; (CAb) animals exposed to IgG isotype control antibody (CAb) on Days 4 and 7, showing 3-D biofilm, bacterial cocci (yellow arrowheads), and host cells (green arrows); (TRL1068) animals exposed to TRL1068 on Days 4 and 7, showing flattened biofilm remnant with no bacteria or host cells present. Note: no antibiotics were given for this experiment. Magnification was the same for all images: width = 75 µm; scale bars: 2 µm.

**Figure 4 antibiotics-12-01490-f004:**
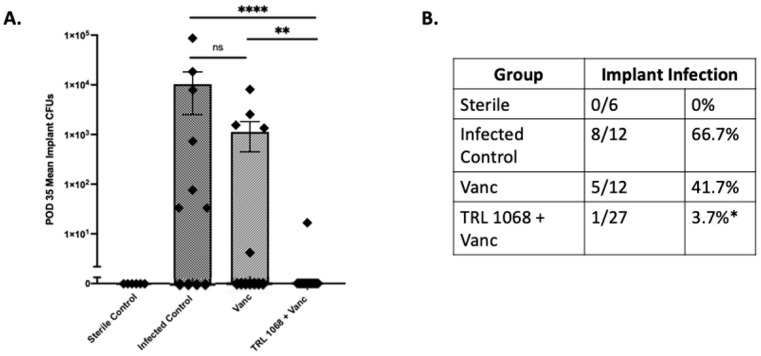
Day 35 implant CFUs. On POD 35, animals were sacrificed, implants were harvested, and CFUs were enumerated. (**A**) POD 35 ex vivo mean implant CFUs. There was a 4-log reduction in average implant CFUs in the TRL1068+vancomycin group relative to the infected control group (6.17 × 10^−1^ vs. 1.03 × 10^4^, respectively; *p* < 0.0001), and a 3.3-log reduction in average implant CFUs in the TRL1068+vancomycin group relative to the vancomycin only group (6.17 × 10^−1^ vs. 1.13 × 10^4^, respectively; *p* = 0.003). (**B**) Implant infection. Pearson’s Chi-square test showed a statistical relationship between treatment and infection status for the implants (χ^2^ [df = 3, N = 78] = 18.73; *p* < 0.001). The TRL1068+Vanc group had significantly lower proportion of infected implants (1/27, 3.7%; *p* < 0.05) compared to the infected control (8/12, 66.7%) and the Vanc (5/12, 41.7%) groups. ns = not significant, * *p* < 0.05, ** *p* < 0.005, **** *p* < 0.0001.

**Figure 5 antibiotics-12-01490-f005:**
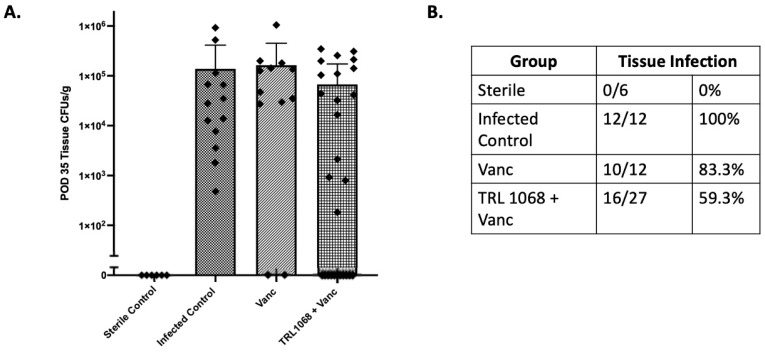
Day 35 peri-implant tissue CFUs. On POD 35, animals were sacrificed, peri-implant tissues were harvested, and CFUs per gram of tissue were enumerated. (**A**) POD 35 ex vivo mean peri-implant tissue CFUs/g. There was no significant difference in the mean peri-implant tissue CFUs/g between any of the experimental groups. (**B**) Pearson’s Chi-square test showed a statistical relationship between treatment and infection status for the implants: χ^2^ [df = 3, N = 78] = 8.30; *p* = 0.03. In the peri-implant tissues, the infected control group had a significantly higher proportion of infected tissues (13/13, 100%; *p* < 0.05) compared to TRL1068+Vanc (16/27, 59.3%) and Vanc (10/12, 83.3%) groups. No significant difference in the proportion of infected tissues was noted among the treatment groups.

## Data Availability

Complete data are available upon request to authors.

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
