# Peer review of "Monoclonal Antibody Disrupts Biofilm Structure and Restores Antibiotic Susceptibility in an Orthopedic Implant Infection Model"

_antibiotics, 2023, doi:10.3390/antibiotics12101490_

Round 1
Reviewer 1 Report
The current manuscript (antibiotics-2608148) presented their investigation on the role of TRL1068, a human monoclonal antibody (mAb) against in an orthopaedic implant infection model and a potential mechanism of action was elucidated. Their data indicated that, the TRL1068 localized to biofilm, in addition, the treatment with TRL1068 avoided the biofilms formation in vivo. Overall, this work is important and interesting to clarify the mechanism of anti-bio effects with mAb for orthopaedic implant. However, some points should be clear before the accept of current work.
1. About the dosage, if it is not too much trouble, the discussion of the dosage of TRL1068 is wanted.
2. As mice were used in current study, an ethics approval number should be offered in Section 4.1.
Author Response
Thank you for your insightful comments. Please see attached cover letter for response.

Reviewer 2 Report
The authors propose an interesting and almost novel paper on possible use of Monoclonal Antibody Disrupts Biofilm Structure and Restores 2
Antibiotic Susceptibility in an Orthopedic Implant Infection 3
Model. This is an intersting and intriguing approach to this kind of infection.
The paper need just minor modifications to be improved
Minor criticism
Please proved reference to Mao Y, Valour F, Nguyen NTQ, Doan TMN, Koelkebeck H, Richardson C, Cheng LI, Sellman BR, Tkaczyk C, Diep BA. Multimechanistic Monoclonal Antibody Combination Targeting Key Staphylococcus aureus Virulence Determinants in a Rabbit Model of Prosthetic Joint Infection. Antimicrob Agents Chemother. 2021 Jun 17;65(7):e0183220 in the introduction and discuss it in discussion. this was a previous paper suggesting this kind of approach but lacking of some of the new approach proposed in this current submitted paper
a table comapring time of receovering/ restoring after the use of monoclonal antibody, if possibile, should be add
minor
Author Response

(The authors gave the same response as above.)
